# Distinct Role of CD11b^+^Ly6G^−^Ly6C^−^ Myeloid-Derived Cells on the Progression of the Primary Tumor and Therapy-Associated Recurrent Brain Tumor

**DOI:** 10.3390/cells9010051

**Published:** 2019-12-24

**Authors:** Sheng-Yan Wu, Chi-Shiun Chiang

**Affiliations:** 1Department of Biomedical Engineering and Environmental Sciences, National Tsing Hua University, 101 Sec. 2, Kuang-Fu Road, Hsinchu 30013, Taiwan; z2743216810@hotmail.com.tw; 2Institute of Nuclear Engineering and Science, National Tsing Hua University, Hsinchu 30013, Taiwan; 3Frontier Research Center on Fundamental and Applied Sciences of Matters, National Tsing Hua University, Hsinchu 30013, Taiwan

**Keywords:** myeloid cells, brain tumor, tumor-associated macrophages

## Abstract

Myeloid-derived cells have been implicated as playing essential roles in cancer therapy, particularly in cancer immunotherapy. Most studies have focused on either CD11b^+^Ly6G^+^Ly6C^+^ granulocytic or polymorphonuclear myeloid-derived suppressor cells (G-MDSCs or PMN-MDSCs) or CD11b^+^Ly6G^−^Ly6C^+^ monocytic MDSCs (M-MDSCs), for which clear roles have been established. On the other hand, CD11b^+^Ly6G^−^Ly6C^−^ myeloid-derived cells (MDCs) have been less well studied. Here, the CD11b-diphtheria toxin receptor (CD11b-DTR) transgenic mouse model was used to evaluate the role of CD11b^+^ myeloid-derived cells in chemotherapy for an orthotopic murine astrocytoma, ALTS1C1. Using this transgenic mouse model, two injections of diphtheria toxin (DT) could effectively deplete CD11b^+^Ly6G^−^Ly6C^−^ MDCs while leaving CD11b^+^Ly6G^+^Ly6C^+^ PMN-MDSCs and CD11b^+^Ly6G^−^Ly6C^+^ M-MDSCs intact. Depletion of CD11b^+^Ly6G^−^Ly6C^−^ MDCs in mice bearing ALTS1C1-tk tumors and receiving ganciclovir (GCV) prolonged the mean survival time for mice from 30.7 to 37.8 days, but not the controls, while the effectiveness of temozolomide was enhanced. Mechanistically, depletion of CD11b^+^Ly6G^−^Ly6C^−^ MDCs blunted therapy-induced increases in tumor-associated macrophages (TAMs) and compromised therapy-elicited angiogenesis. Collectively, our findings suggest that CD11b^+^Ly6G^−^Ly6C^−^ MDCs could be manipulated to enhance the efficacy of chemotherapy for brain tumors. However, our study also cautions that the timing of any MDC manipulation may be critical to achieve the best therapeutic result.

## 1. Introduction

Gliomas are the most common primary intracranial brain tumors, accounting for 81% of malignant brain tumors [1,2]. Despite improvements in medical treatments, the survival time for patients has remained very short, averaging 14.6 months for newly diagnosed patients and only five months after recurrence [3,4,5]. One factor in the minimal improvement in survival may be therapy-induced myelopoiesis, which leads to the infiltration of myeloid cells into the tumor from the peripheral blood and a high content of tumor-associated macrophages (TAMs). TAMs are most often of the M2 phenotype that, along with myeloid-derived suppressor cells (MDSCs), promote rapid tumor proliferation and create a microenvironment that supports tumor survival by stimulating angiogenesis, promoting tumor invasion, and suppressing the anti-tumor immunity [6,7].

The myeloid cells have gained in notoriety since the early 1990s, when researchers found that immature CD11b^+^Gr-1^+^ myeloid cells were phenotypically similar to neutrophils and monocytes in the peripheral blood, but differed in functionality. The most obvious difference was their ability to suppress immune responses, leading to the introduction of the term MDSCs since 2007 [8,9,10]. Since Gr-1 is a composite epitope present in both Ly6G and Ly6C molecules that is differentially expressed by a sub-population of the myeloid cell, scientists have further divided MDSCs into two subgroups: either CD11b^+^Ly6G^+^Ly6C^+^ granulocytic or polymorphonuclear myeloid-derived suppressor cells (G-MDSCs or PMN-MDSCs) or CD11b^+^Ly6G^−^Ly6C^+^ monocytic MDSCs (M-MDSCs) [11,12]. Growing evidence has demonstrated the effects of MDSCs on tumor progression, metastasis, and immune evasion in various tumor models including gliomas [13,14]. On the other hand, CD11b^+^Ly6G^−^Ly6C^−^ myeloid-derived cells (MDCs) in the blood have been less studied, even though they are the precursors of all myeloid cells including MDSCs [10,15].

Myeloid cells have long been regarded as the first line of defense for the immune system. Recently, various reports have shown that malignant tumors can release chemokines like monocyte chemoattractant protein-1 (MCP-1/CCL2) and macrophage inflammatory protein 2-alpha (MIP-2α/CXCL2), which attracts CD11b^+^ myeloid cells from blood into the tumor microenvironment where they differentiate into tumor-associated macrophages (TAMs) that are most often of the M2 phenotype that expresses high levels of anti-inflammatory factors such as IL-10, IL-4, and MMP9 that help tumor progression and vascularization in various types of tumors including breast, pancreatic, prostate, lung, and gastric cancers [16,17,18,19,20]. Gliomas also have a high proportion of myeloid cells, with 30% to 50% being macrophages/microglia [21,22,23,24]. Myeloid cells in the tumor microenvironment are a potential target for adjuvant therapy, but their heterogeneity makes analysis of the roles of different phenotypes challenging.

Genetic targeting is frequently used to analyze many biological functions or processes. Researchers have used CD11b-DTR transgenic mice expressing the human diphtheria receptor (DTR) under the control of the human integrin alpha M promoter (ITGAM/CD11b) to deplete CD11b^+^ cells in mice following the administration of the diphtheria toxin (DT) [25,26,27]. This model has been used to deplete CD11b cells in peritoneal macrophages, Gr-1^+^ myeloid populations, and TAMs in pancreatic, colon, and liver cancers [28,29,30,31,32,33]. The degree of CD11b^+^ cell depletion varied in the reports, which were mainly the results of the administration protocol of the DT agent [29,30,31,32,33]. Our study, using this transgenic mouse model with two injections of the medium dose of DT, found that CD11b^+^Gr-1^−^, but not CD11b^+^Gr1^+^ cells, were depleted from the brain tumor. Suicide gene therapy was another genetic targeting approach first proposed by Moolten in 1986 [34]. The overexpression of herpes simplex virus type-1/thymidine kinase (HSV-tk) with the administration of pro-drug ganciclovir (GCV) is a standard model in preclinical and clinical studies for studying pro-drug therapy [35,36]. HSV-tk is 1000 times more efficient than the mammalian thymidine kinase to use monophosphorylate GCV (thymidine analog to inhibit DNA synthesis) as a substrate. By expressing HSV-tk in the cancerous cells, selective cytotoxicity against tumor cells is therefore achieved [37]. These genetic targeting methods allowed us to investigate the role of the CD11b^+^Gr-1^−^ myeloid cells in brain tumor progression and recurrence after pro-drug GCV chemotherapy in our established astrocytoma model.

We have developed a murine astrocytoma cell line, ALTS1C1, that has a heavy myeloid cell infiltrate and resembles glioblastoma in its aggressive features [38]. Even local radiation therapy (RT) in this model stimulates myelopoiesis and modifies the tumor microenvironment in favor of anti-inflammatory M2 TAMs, which promote tumor recurrence [39]. Interestingly, in another model of tk/GCV gene therapy combined with IL-3 immune therapy in a prostate tumor model, M1 macrophages were generated that had anti-tumor efficacy with NO production [37]. Our aim here was to specify the roles of CD11b^+^Gr-1^−^ myeloid cells in ALTS1C1 tumors following chemotherapy. We demonstrated that the CD11b-DTR mice model is a suitable macrophage ablation model for ALTS1C1 brain tumor research and that F4/80^+^ macrophages and Ly6C^−^ monocytes were selectively depleted, which enhanced the chemo-therapeutic effect while depletion without chemotherapy did not slow tumor progression. Our findings also revealed the importance of the timing of treatments. These discoveries may provide insights into future myeloid cell targeting adjuvant therapies for brain tumors.

## 2. Materials and Methods

### 2.1. Mice

C57BL/6J mice aged 6–8 weeks old were purchased from the National Laboratory Animal Center of Taiwan. The CD11b-DTR transgenic mice were from The Jackson Laboratory (The Jackson Laboratory, 006000, USA). All experiments and animal handling were conducted according to the guidelines under the approval of the Institutional Animal Care and Use Committee of National Tsing Hua University (IACUC protocol No.: 10419), Taiwan.

### 2.2. Cell Line Cultures

The murine astrocytoma cell line, ALTS1C1 (BCRC60582, BCRC, Hsinchu, Taiwan; T8239, Applied Biological Materials, BC, Canada), was previously established by our laboratory [38]. ALTS1C1 cells were incubated at 37 °C/5% CO_2_ humidified air condition and maintained in the culture medium. Culture medium was prepared in Dulbecco’s modified Eagle’s medium (DMEM; Gibco^®^, 12100046, Grand Island, NY, USA) with 10% fetal bovine serum (FBS; Gibco^®^, 16000044) and 1% penicillin-streptomycin (PS; Gibco^®^, 15140122). Mycoplasma contamination was examined by a EZ-PCR™ Mycoplasma Detection Kit (Biological Industries, 20-700-20, Beit Haemek, Israel) before use.

### 2.3. Plasmids and Cell Transfection

The HSV-sr39tk suicide gene constructed by Dr. Ching-Fang Yu [37] was transfected into ALTS1C1 by Effectene Transfection Reagent (QIAGEN, 1054250, Mainz, Germany). Briefly, 1 μg of plasmid DNA diluted to a total of 150 μL and mixed with 8 μL Enhancer and kept at room temperature for 4 min. A total of 25 μL of Effectene Transfection Reagent was mixed and kept at room temperature for another 8 min. The final culture medium was replaced with plasmid containing a medium mixture. Two days later, transfected cells were selected by G418 (Promega, 216436, Madison, WI, USA) 1.5 mg/mL in culture medium for 10–14 days. The stably transfected cells were named ALTS1C1-tk.

### 2.4. Flow Cytometry Analysis

Peritoneal cells and systemic blood cells were collected for flow cytometry analysis. Briefly, mice were first anesthetized and followed by peritoneal lavage with 5 mL cold PBS (Biological Industries, 02-023-5A) in the use of a 23G needle (TOP). Later, mice were shaken three times, and the peritoneal cell suspension was collected. Blood cells were collected using 0.01% EDTA (SIGMA, 6381-92-6, Saint Louis, MO, USA) rinsed animal lancet (Goldenrod, GR 4 mm, Braintree, MA, USA) from the mice cheek. 1× RBC (Red Blood cell) lysis (eBioscience, 00-4300-54, Carlsbad, CA, USA) was added to lyse red blood cells for 5 min. PBS was applied to stop the RBC lysis reaction and the cell suspension was collected. Cell suspensions were blocked with 1% goat serum (Gibco^®^, 16210-064) and 0.2% FC block (BD Pharmingen, 553142, Franklin Lakes, NJ, USA) for 30 min. After blocking, cells were stained with fluorescence conjugated antibody against CD11b, Ly6C, Ly6G, or CD45 (BD Phamingen 550993, 553104, 551461, 552848, respectively), and F4/80 (Bio-RAD, MCA497APC, Kidlington, UK) for 1 h. The cell suspensions were washed by PBS twice before FACS (Fluorescence-activated cell sorting) analysis on Canto cytometer^TM^ (BD Bioscience, 337175) and data were analyzed by FACSDiva software v6.1.3 (BD Pharmingen).

### 2.5. Orthotopic Intracranial Tumor Model

To establish intracranial tumors, ALTS1C1 orALTS1C1-tk cells were inoculated into the brain of 8–12 weeks old C57BL/6J or CD11b-DTR mice with the procedures described in a previous publication [38]. Briefly, mice were anesthetized, and 1 × 10^5^ cells in 2 μL were intracranially (i.c.) injected at 0.1 mm posterior to the bregma and 2.0 mm laterals to the midline with a 2.5 mm depth. After injection, the hole was sealed with bone wax (ETHICON, W810, Somerville, NJ, USA), and two stitches were performed to suture the skin of the mice. When neurologic deficits (lethargy, failure to ambulate, and loss of more than 20% of original body weight) showed, mice were sacrificed. Brain tissue with a tumor inside the skull was carefully removed, embedded in OCT (Optimal Cutting Temperature) compound (Sakura Finetek, 4583, Torrance, CA, USA), and stored in −80 °C refrigerator. Hematoxylin (SIGMA, GHS232-1L) and eosin (SIGMA, HT110116-500ML) staining were performed to analyze the tumor’s largest section area as well as histopathology.

### 2.6. Immunohistochemistry

Frozen tissues were sectioned (10 μm), mounted onto slides, and stored at −20 °C. For immunohistochemistry analysis, section slides were fixed with methanol and permeabilized with 0.05% Tween-20 (SIGMA, p1379-500ML). The slides were subsequently mounted with blocking buffer (4% FBS and 1% goat serum in 1 × PBS) for 1 h to prevent non-specific binding. After blocking, the first antibodies were used as follows: rat anti-mouse CD11b (BD, 550282), rat anti-mouse F4/80 (Serotec, MCA497GA), rat anti-mouse GR-1 (BD, 553123), rat anti-mouse CD31 (BD, 550274), mouse anti-mouse ARG-1 (BD, 610708), and rabbit anti-mouse iNOS (Millipore, 2281700, Burlington, MA, USA) with a one to 200 dilution in blocking buffer. The slides were stained overnight in 4 °C and then washed with two rinses of PBS. Indicated specific host secondary antibodies conjugated with Alexa Fluor 488 (Invitrogen, A11008, A21121) or Alexa Fluor 594 (Invitrogen, A11007, A21125) were mounted for one hour. Slides were then washed with two rinses of PBS and then mounted with DAPI (4′,6-diamidino-2-phenylindole) (Invitrogen, P36931) for nucleus visualization. Images were taken by the AxioCam MRC-5 camera on an Axiovertskop 40 fluorescence microscope (Carl Zeiss, Jena, Germany) and analyzed by Image-Pro Plus 6.0, ImageJ 1.48v software. Quantification of myeloid cell percentage was calculated as the proportion of CD11b-, F4/80- and GR-1-positive cells in the tumor region (condensed dapi positive area). Microvascular density (MVD) was calculated as the percentage of CD31 positive area in the tumor region (condensed dapi positive). 

### 2.7. Drug Preparation

A total of 100 mg GCV powder (SIGMA, G2536-100MG) was dissolved in 10 mL methanol (40 μm strainer filtered) and stored at 4 °C as the stock solution. GCV was given to mice with intraperitoneal (i.p.) injection twice a day as 10 mg/kg mice body weight for consecutive six days. DT (SIGMA, D0564-1MG) was freshly diluted to 2.5 μg/mL in PBS and mice were i.p. injected with DT as 10 μg/kg mice body weight for consecutive two days. A total of 1 mg of TMZ (Temozolomide) powder (SIGMA, T2577-25MG) was freshly dissolved in 63 μL DMSO (J.T. Baker, 9224-03, Center Valley, PA, USA) and diluted to 20% with 1× PBS. Mice were i.p. injected with TMZ as 50 mg/kg mice body weight for six consecutive days.

### 2.8. Statistics

Experiments were performed in at least three repeats with one assay for analysis. Statistics were performed using the two-tailed Student’s t-test or one-way ANOVA by GraphPad Prism 5 (San Diego, CA, USA). A *p* value ≤ 0.05 was regarded as statistically significance.

## 3. Results

### 3.1. Selective Myeloid Cells Depletion in CD11b-DTR Transgenic Mice

To confirm myeloid cell depletion in CD11b-DTR transgenic mice, two injections of DT were used. Peritoneal cells and white blood cells were examined by flow cytometry at the indicated time (Figure 1A). The results confirmed the report by [29] that CD11b^+^F4/80^+^ peritoneal macrophages could be significantly reduced following DT administration compared to the PBS-treated group (1.06% vs. 8.62%, 3.26% vs. 10.26%, at day three and day six, respectively) (Figure 1B). Cells from red cell lysed blood were gated first by the CD11b positive area and then by Ly6C and Ly6G to give three distinct groups (Appendix A). The analysis showed that CD11b^+^Ly6G^−^Ly6C^−^ MDCs were the most decreased after DT treatment (Figure 1C), but the CD11b^+^ Ly6G^+^Ly6C^+^ PMN-MDSCs (Figure 1D) and CD11b^+^Ly6G^−^Ly6C^+^ M-MDSCs (Figure 1E,F) were not significantly affected and indeed were increased despite all expressing CD11b. The above data concluded that the DT dose used in this study could transiently deplete peritoneal macrophages and systemic MDCs, but not PMN-MDSCs and M-MDSCs in the CD11b-DTR mouse in our protocol.

### 3.2. Depletion of the CD11b^+^Ly6G^−^Ly6C^−^ MDCs Alone Did Not Impede Tumor Growth

To evaluate the roles of the CD11b^+^Ly6G^−^Ly6C^−^ MDCs in brain tumor progression, we used an established ALTS1C1-based orthotopic astrocytoma tumor model [38] in CD11b-DTR transgenic mice, with two injections of DT given 11 days post tumor inoculation (Figure 2A). Blood samples taken at specific time points showed a similar trend to previous data where CD11b^+^Ly6G^−^Ly6C^−^ MDCs (Appendix AA) were significantly decreased after DT administration. The PMN-MDSCs and the M-MDSCs in tumor-bearing mice were unaffected and did not increase, unlike in the controls (Appendix AB,C). The survival data revealed that 50% of tumor-bearing mice died around 25.0 ± 1.9 days after intracranial injection of ALTS1C1 tumor cells without any treatment. The decrease in CD11b^+^Ly6G^−^Ly6C^−^ MDCs did not prolong the life of mice bearing ALTS1C1 tumors, in fact, the mice died earlier (mean surviving day = 21.3 ± 3.3 days) than the control group (Figure 2B). One criterion for survival analysis is the bodyweight loss, and we noticed that the DT-treated mice had less appetite than the control mice, which may partially explain why the DT-treated mice had shorter mean surviving times. A more detailed investigation of tumor progression was performed on tumor-bearing mice sacrificed at the time to quantify the tumor section area by H&E staining (Figure 2C), but there was no significant difference in tumor size regardless of whether MDCs were depleted or not (Figure 2D). This result revealed that the depletion of MDCs alone did not affect tumor growth.

To better visualize myeloid cells in the tumor microenvironment, brain tumor tissues were examined by IHC (immunohistochemistry) staining (Figure 3A). The results revealed that the DT treatment only slightly reduced the total number of CD11b myeloid cells in the tumor (Figure 3B) with decreases only in the number of F4/80^+^ TAMs (Figure 3C), but not in Gr-1^+^ granulocytes (Figure 3D). We also stained for the endothelial marker CD31 and found no changes in the mean vessel density (MVD) of the primary tumor (Figure 3E).

### 3.3. Selective MDCs Depletion Benefits tk/GCV Therapy

Although depletion of CD11b^+^Ly6G^−^Ly6C^−^ MDCs and F4/80^+^ TAMs seemed to have little effect on tumor progression, we determined whether they would have roles in response to therapeutic drugs as frontline chemo-drug treatment or radiation therapies promote myeloid cells [40,41,42,43,44]. To achieve this goal, we established a tk/GCV therapy system using this cancer model (Appendix A). The DT was given one to two days after GCV treatment, which was given twice a day for six consecutive days beginning ten days after tumor inoculation (Figure 4A). The mean survival time for ALTS1C1-tk tumor-bearing mice increased from 25 days (PBS control group) (Figure 2B) and 30.7 days (GCV treatment only group) (Figure 4B) to 37.8 (DT combined GCV treatment group) days (Figure 4B). When the mice were terminated at day 43, one mouse from the DT and GCV treatment group was completed cured with no detectable tumor residues in the brain (data not shown). The analysis of the tumor section measurement (Figure 4C) confirmed that depletion of CD11b^+^Ly6G^−^Ly6C^−^ MDCs enhanced the cytotoxicity of GCV against ALTS1C1-tk tumors, as shown by the smaller tumor size on day 16 (Figure 4D). This analysis also showed that mice died at different times between the GCV (~28 days) and GCV + DT (~32 days) groups (Figure 4C), but with a similar size (Figure 4D). This indicates that the mice mainly died from the tumor pressure and not from other reasons.

IHC staining for myeloid cells and blood vessels in tumor tissues (Figure 5A) revealed that tk/GCV treatment significantly increased the number of infiltrating CD11b^+^ myeloid cells (32.1% to 61.4%), and F4/80^+^ TAMs (15.0% to 36.1%), but not Gr-1^+^ granulocytes (6.9% to 3.6%) (Figure 5B–D) when compared with the control group (Figure 3). The CD31 staining showed that the MVD also increased from 4.1% to 7.4% (Figure 5E). In contrast, DT treatment blunted the tk/GCV treatment-elicited increase in CD11b^+^ myeloid cells (61.4% to 17.0%) (Figure 5B) and F4/80^+^ TAMs (36.1% to 3.6%) (Figure 5C), while Gr-1 granulocytes were unaffected (Figure 5D). The MVD also decreased after myeloid cell depletion to a similar level as the controls (reduced from 7.4% to 4.4%, comparable to the PBS tumor). Taken together, the data indicated that depletion of CD11b^+^Ly6G^-^Ly6C^-^ MDCs and F4/80^+^ TAMs selectively compromised tk/GCV therapy and their depletion enhanced the efficacy of tk/GCV treatment to prolong mouse survival.

Next, we investigated whether CD11b^+^Ly6G^−^Ly6C^−^ MDCs depletion could enhance the efficacy of the current standard glioma chemotherapeutic agent Temozolamide (TMZ) [5]. TMZ was given to mice ten days after tumor inoculation and once a day for six consecutive days. DT was administrated during the TMZ treatment (Figure 5F). H&E staining indicated that the depletion of macrophages enhanced TMZ-induced shrinkage of the tumor compared to TMZ alone or the control group (Figure 5G).

### 3.4. The Time-Dependent Effect of DT Administration

We have demonstrated that the selective depletion of CD11b^+^Ly6G^−^Ly6C^−^ MDCs and F4/80^+^ TAMs strengthened the efficacy of tk/GCV therapy. To explore whether the benefit was time-dependent, we depleted myeloid cells either before GCV therapy (DT was given at days 8 and 9 after i.c. injection) or after GCV therapy (DT was given at days 16 and 17 after i.c. injection) (Figure 6A). There was no significant difference in survival when DT was given before or after the GCV treatment compared to GCV treatment alone (mean 34.0 vs. 34.3 vs. 31.3 days, respectively) (Figure 6B). The most significant effect was when DT was given together with tk/GCV treatment, so CD11b^+^Ly6G^−^Ly6C^−^ MDCs depletion was time-dependent.

## 4. Discussion

Evidence suggests that TAMs can promote tumor proliferation and angiogenesis and that this can be enhanced by some cancer therapies such as paclitaxel, doxorubicin, and platinum, leading to greater macrophage infiltration into tumors that may promote tumor recurrence [45,46,47,48]. In the present study, we show that CD11b^+^F4/80^+^ macrophages massively increased after tk/GCV chemotherapy in the brain tumor-bearing mice. To examine the role of these macrophages on tk/GCV chemotherapy, we took advantage of the CD11b-DTR transgenic mice to achieve selective ablation of CD11b^+^Ly6G^−^Ly6C^−^ MDCs and F4/80^+^ TAMs, but not PMN-MDSCs and M-MDSCs. Ablating the tk/GCV therapy-induced increase in these macrophage subsets greatly enhanced treatment efficacy and animal survival, even though there was no effect on tumor progression in untreated controls. These findings indicate that CD11b^+^Ly6G^−^Ly6C^−^ MDCs and F4/80^+^ TAMs might have different roles in primary and chemotherapy-treated tumors.

The role of TAMs in promoting tumor angiogenesis has been widely reported [40,49,50,51] and in keeping with this, we found that the MVD was significantly increased after tk/GCV treatment, which was reversed by macrophage depletion in CD11b-DTR mice. These findings suggest that CD11b^+^Ly6G^−^Ly6C^−^ are the primary source for the increased TAMs after therapy and are responsible for therapy-induced angiogenesis and tumor progression. Presumably, because the depletion of TAMs before or after tk/GCV therapy had no significant impact on the overall survival, therapy-induced TAMs must act to promote tumor recurrence during the time when therapy is being administered. This could be a time issue as macrophage depletion in this model is transient. It is likely that the transient decrease of TAMs has an essential role in the initial regrowth of drug treated-tumor, but not thereafter [52,53]. The time-dependent factor may also explain why other successful depletion studies were performed at a very early stage after tumor inoculation [31,54,55]. However, we cannot ignore a possible compensatory effect of the increase of PMN-MDSCs and M-MDSCs where macrophages are depleted in the absence of chemotherapy.

It needs reminding that the myeloid cells that we have described as depleted here only include the CD11b^+^Ly6G^−^Ly6C^−^ population in the blood, and peritoneal and intratumoral CD11b^+^F4/80^+^ cells. Since microglia are one of the primary myeloid cells in the brain and an alternative source of pro-angiogenic factors [7,56], we attempted to verify whether resident microglia were deleted in CD11b-DTR transgenic mice. The literature reports conflict on this point. Frieler and his team discovered that microglia in the cortex and subcortex were unaffected [57]; however, Ueno et al. reported that one-third of microglia in the subcortical white matter were reduced 12 h after DT injection, though they had recovered 24 h later [58]. In our model, flow cytometry on brain cells using CD11b^+^CD45^low^ as a marker of microglia showed no depletion in the brains of CD11b-DTR mice after the two DT injections (data not shown). The differences between these models may be due to the daily DT dose used or disease model. The studies showing depletion used over five times the dose that we used. Therefore, we do not think that microglia were relevant players in our system, although this needs to be confirmed, preferably using transmembrane protein 119 (Tmem119) and purinergic receptor (P2ry12) as markers to identify brain microglia [59,60].

While TAMs are the most abundant myeloid cells in tumor, granulocytes or neutrophils are the most abundant in blood. They are the first responders to the acute inflammation and tissue damage in the host body [61,62,63]. It has been reported that granulocytes are not affected in this CD11b-DTR mouse model, although they are CD11b positive. That the CD11b^+^Gr1^+^ cells did not respond to the cytotoxicity of DT in this mouse model is likely to be an issue of dose tolerance. Each subtype of CD11b^+^ cells has a different expression level of CD11b and, therefore, the tolerance to DT. This variation may also explain why reports showed a discrepancy on types of cell depletion in using this mouse model [28,29,30,31,32,33] as some have said that they transiently increase 24 h after DT administration [29,57,64], which may represent emergency mobilization. Our findings also revealed an increase in the percentage of CD11b^+^Ly6G^+^Ly6C^+^ granulocytes (or PMN-MDSCs) and CD11b^+^Ly6G^−^Ly6C^+^ monocytes (or M-MDSCs) after DT injections (Figure 1D), but not in tumor-bearing CD11b-DTR mice (Figure 3D). On the other hand, tumor-bearing mice had twice the number of granulocytes than the controls ten days after tumor inoculation, which may have affected any response. Overall, we think that neither the systemic granulocytes nor intra-tumoral Gr-1 cells changed significantly throughout the treatment.

TAMs, however, are clearly major players in the tumor microenvironment and targeting these cells as an adjuvant cancer therapy has long been considered a promising clinical approach. Anti-CSF1R and anti-CCL2 blocking antibodies in combination with chemotherapy are in phase I/II clinical trials in pancreatic, breast, and liposarcoma cancers [65,66,67]. Our data suggest that gliomas might also be a good disease for macrophage targeting as an adjuvant cancer therapy.

In summary, our results show that peripheral CD11b^+^Ly6G^−^Ly6C^−^ MDCs and tumor F4/80^+^ TAMs play important roles in decreasing the efficacy of cancer therapies and targeting them may improve outcomes. The timing is best when myeloid depletion is concurrent with chemotherapy. This discovery hints to future pre-clinical and clinical investigations for combining monocyte/macrophage targeting with conventional cancer therapies.

## Figures and Tables

**Figure 1 cells-09-00051-f001:**
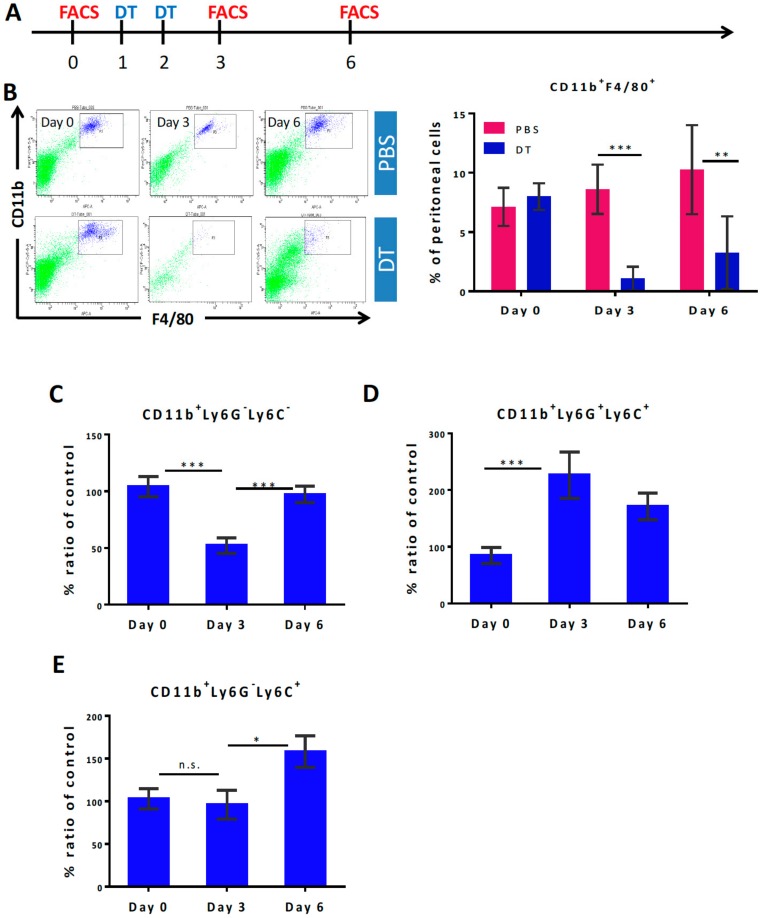
Selective myeloid cell depletion in transgenic CD11b-DTR mice (**A**) The timeline of DT administration and flow cytometry analysis (FACS) on lysed blood cells. (**B**) Representative flow cytometry images of peritoneal cells gated by CD11b and F4/80 and the change in percentage of CD11b^+^F4/80^+^ peritoneal macrophages following the DT treatment at the indicated times (*n* ≥ 3 for each group). (**C**–**E**) Changes in myeloid subgroups in lysed blood stained with Ly6G and Ly6C antibodies and analyzed by flow cytometry (*n* ≥ 10 for each group). Statistics were performed using one-way ANOVA by GraphPad Prism 5. *, *p* < 0.05; **, *p* < 0.01; ***, *p* < 0.001; n.s., *p* > 0.05.

**Figure 2 cells-09-00051-f002:**
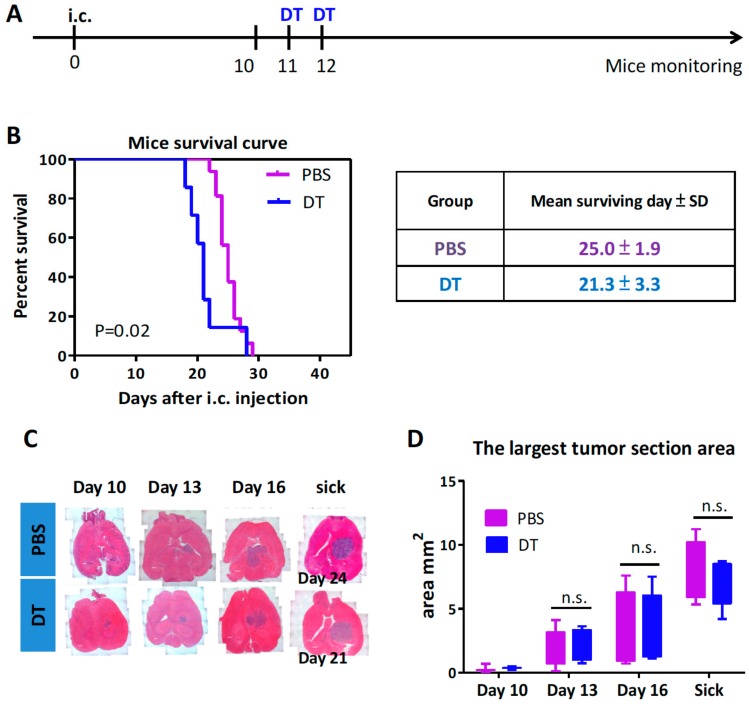
Depletion of CD11b^+^Ly6G^−^Ly6C^−^ MDCs alone did not impede tumor growth. (**A**) The timeline of in vivo DT treatment of CD11b-DTR mice. (**B**) Kaplan–Meier survival curves of ALTS1C1 tumor-bearing mice receiving DT (PBS group *n* = 16, DT group *n* = 7). (**C**) Representative H&E stained brain tissue at different time points after tumor inoculation or when mice became sick. (**D**) Measurement of the largest tumor section area at different times after tumor inoculation or when mice became sick (*n* ≥ 3 for each group).

**Figure 3 cells-09-00051-f003:**
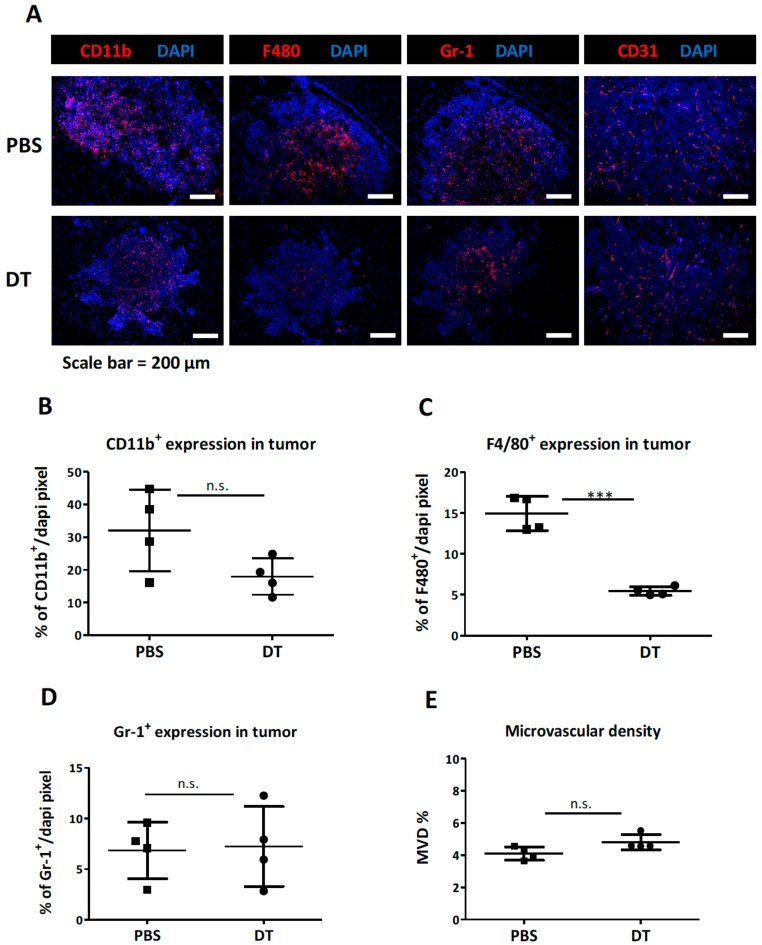
The change in tumor microenvironment following the DT administration in CD11b-DTR mice-bearing ALTS1C1 tumors. (**A**) Representative pictures of tumor sections stained with DAPI and myeloid cell markers, CD11b (red), F4/80 (red), and Gr-1 (red) as well as an endothelial marker, CD31 (red). The nucleus was stained with DAPI (blue) (**B**–**D**) Quantitative data of IHC staining for CD11b, F4/80, and Gr-1, on CD11b-DTR mice-bearing ALTS1C1 tumors at day 13 (one day after mice received two DT injections). (**E**) MVD (mean vessel density) is shown as the percentage of CD31 positive area in the DAPI-positive tumor area. Symbols of one dot indicates one mouse, and the error bars are mean with ± S.D. Each group had at least three mice. Scale bar = 200 μm. ***, *p* < 0.001.

**Figure 4 cells-09-00051-f004:**
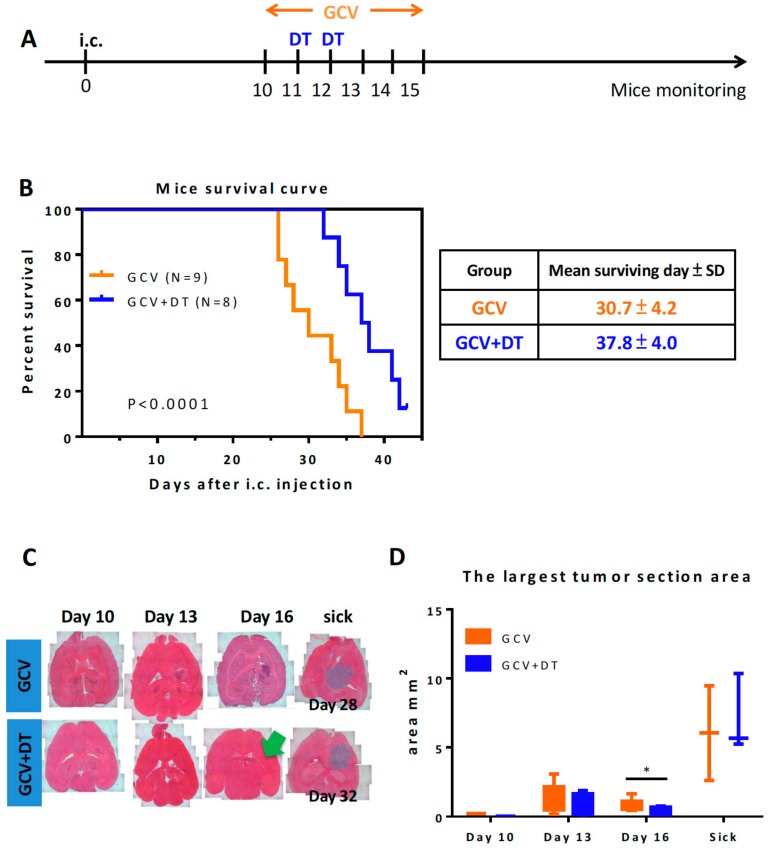
Selective CD11b^+^Ly6G^-^Ly6C^-^ MDCs depletion benefits tk/GCV therapy. (**A**) The timeline of in vivo DT and tk/GCV treatment on CD11b-DTR mice-bearing ALTS1C1-tk tumors. (**B**) Kaplan–Meier survival curves of ALTS1C1-tk tumor-bearing mice receiving GCV only or combined DT and GCV treatments (GCV group *n* = 9, GCV + DT group *n* = 8), *p* < 0.0001. (**C**) Representative H&E stained brain tissues at different time points after tumor inoculation or the time when mice became sick. (**D**) Measurement of the largest tumor section area at different times after tumor inoculation or when mice became sick (*n* ≥ 3 for each group). *, *p* (0.03) < 0.05 at day 16.

**Figure 5 cells-09-00051-f005:**
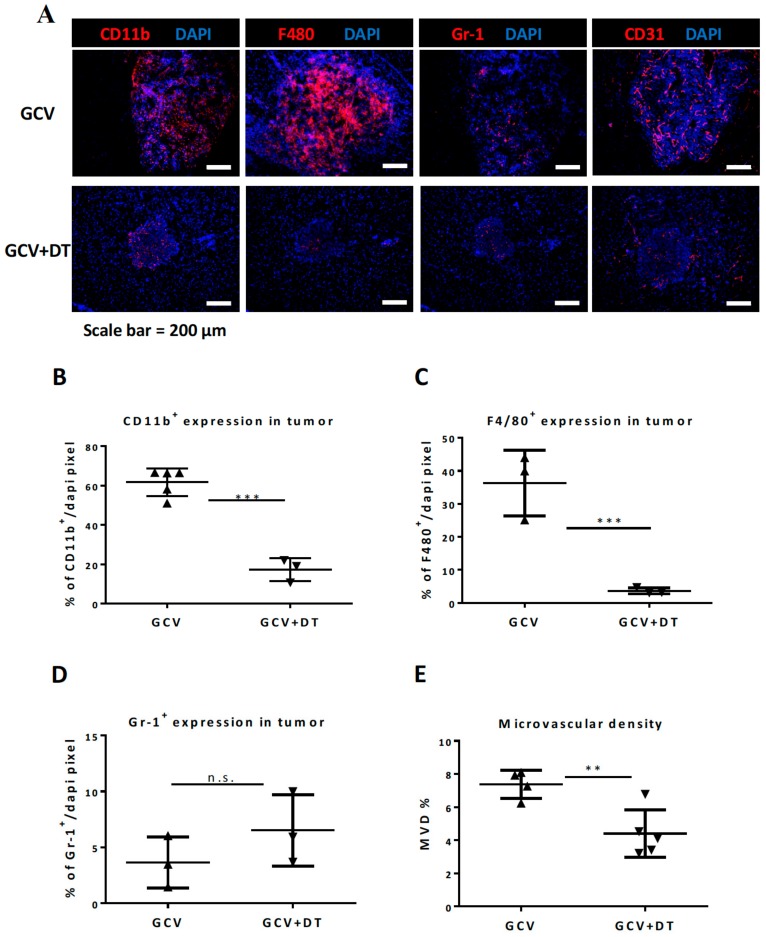
CD11b^+^Ly6G^−^Ly6C^−^ MDCs depletion blocked the increase in ctk/GCV therapy-induced TAMs. (**A**) Representative pictures of tumor sections stained with DAPI and myeloid cell markers. (**B**–**E**) Quantitative data of day 13 IHC staining of ALTS1C1-tk tumors receiving tk/GCV or combined with DT treatment in CD11b-DTR mice. The myeloid cells were stained with CD11b (red), GR-1 (red), F4/80 (red), vessels were stained with CD31 (red), and the nucleus was stained with DAPI (blue). MVD (mean vessel density) is shown as the percentage of CD31 positive area in the DAPI-positive tumor area. Symbols of one dot indicate one mouse, and the error bars are mean with ± S.D. Each group had at least 3 mice. Scale bar = 200 μm. **, *p* < 0.01; ***, *p* < 0.001. (**F**) Scheme of DT treatment and TMZ therapy on CD11b-DTR mice-bearing ALTS1C1 tumor. (**G**) H&E staining of day 16 of brain tissue was performed to quantify the largest tumor section area.

**Figure 6 cells-09-00051-f006:**
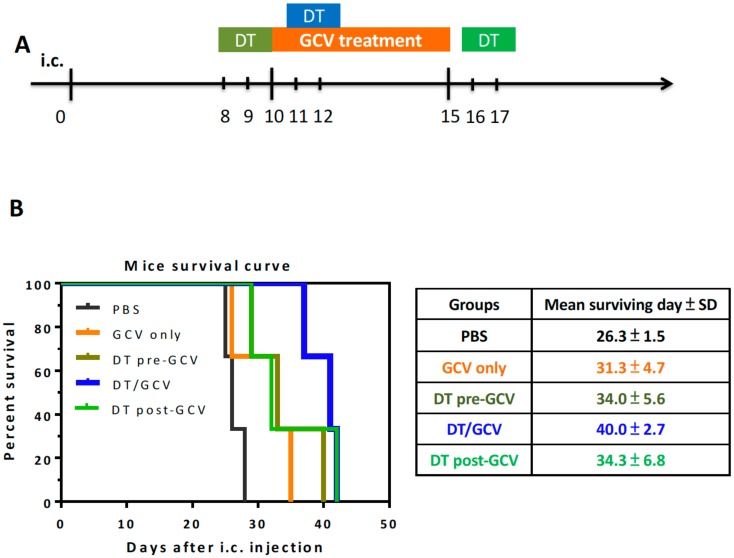
The time-dependent effect of myeloid cell depletion. (**A**) The timeline of DT treatment and tk/GCV therapy in ALTS1C1-tk bearing CD11b-DTR mice. GCV was given twice a day from day 10 to day 15. DT/Pre-GCV: DT was given once a day at days 8 and 9. DT/GCV: DT was given once a day at days 11 and 12. DT/Post-GCV: DT was given once a day at days 16 and 17. (**B**) Kaplan–Meier survival curves of ALTS1C1-tk tumor-bearing mice receiving GCV combined with different DT administration timeline (Each group mice *n* = 3).

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
