# Peer review of "Distinct Role of CD11b+Ly6GLy6C Myeloid-Derived Cells on the Progression of the Primary Tumor and Therapy-Associated Recurrent Brain Tumor"

_cells, 2019, doi:10.3390/cells9010051_

Round 1

Reviewer 1 Report

Revision Wu et al.:

„Distinct role of CD11b+Ly6G-Ly6C- myeloid-derived cells on the progression of the primary tumor and therapy-associated recurrent brain tumor”

The authors describe that CD11b+Ly6G-Ly6C- myeloid-derived cell depletion work synergistically with tk/GCV therapy in a time-dependent manner. Moreover, the study highlights the tumor-promoting impact of tumor-associated macrophages in gliomas.

Major remarks:

1.) Proofread English language, especially the abstract bears several mistakes, e.g. line 16 “have a much clear role of” does not make sense. There is a huge number of typos, wrong expressions, and flawed sentences throughout the manuscript. The overall language in the discussion part appears like spoken language (slang-like) and has to be improved strongly before publication. I suggest that a native speaker should edit the whole manuscript before resubmission.

2.) It appears that DT treatment selectively kills CD11b+Ly6G-Ly6C- cells but no other CD11b positive cell types, which are reported to rather increase upon DT treatment. How do the authors explain this selectivity of DT and the increase if DTR is expressed under the control of the CD11b promoter?

3.) Fig 2B clearly shows that tumor-bearing mice die significantly earlier (p<0.02) when DT is applied. However, the authors claim that this effect is not significant when more mice are pooled for analysis and refer to Fig S3D at that point. Why do the author not show the whole number of mice analyzed in Fig. 2B, but only a subset?

It is also not clear how Fig S3D relates to Fig 2B as no DT-curves are shown in comparison to PBS-curves in Fig S3D as they are presented in Fig 2B. Instead GCV curves are compared to PBS curves. Thus, the data in Fig S3D do not appear to support the claim that “this effect is not significant when more mice are pooled”. Please clarify this and clearly show only relevant data supporting your claim.  

4.) The authors say that “DT treatment slightly reduced the total amount of CD11b cells myeloid cells” which is visible in Fig. 3A but presented as not significant in Fig. 3B. Can the authors increase the number of samples tested (higher n) to reach significance?

5.) Please introduce and explain the tk/GCV therapy system. No reader who is not a particular expert in your field knows what it is. It is impossible to evaluate the data, if it is not even clear how the main experimental system works. It has to be explained in the main text, Fig S3 is not sufficient for that.

6.) In the discussion the authors mention that clinical inhibition of CSF1R and/or CCL2 is our investigation in other tumor types. Is it possible to test inhibitors of these targets in the here employed mouse model to confirm the findings of this study using another method and to present a more specific therapeutic indication form glioma treatment?  

Minor remarks:

1.) The abbreviation GCV is explained or introduced at no point in the manuscript. Please introduce this compound to the readers.

2.) Is it necessary to show in Fig. 4D that sick mice of both populations have similar tumor sizes?

3.) Text within the figures is frequently too small to read in a printout, e.g. Figure 5B.

Author Response

Response to Reviewer 1 comments

Major remarks:

1.) Proofread English language, especially the abstract bears several mistakes, e.g. line 16 “have a much clear role of” does not make sense. There is a huge number of typos, wrong expressions, and flawed sentences throughout the manuscript. The overall language in the discussion part appears like spoken language (slang-like) and has to be improved strongly before publication. I suggest that a native speaker should edit the whole manuscript before resubmission.

Response 1: Prof. Willam H. McBride from UCLA has proofread the manuscript.  We have added his contribution to the acknowledgment section. Many incorrect sentences have been rewritten. Hopefully, the clarity of this manuscript has been improved.

2.) It appears that DT treatment selectively kills CD11b+Ly6G-Ly6C- cells but no other CD11b positive cell types, which are reported to rather increase upon DT treatment. How do the authors explain this selectivity of DT and the increase if DTR is expressed under the control of the CD11b promoter?

Response 2: We considered this could be an issue of DT dose and treatment protocol.  The increase of CD11b+Ly6G+Ly6C+ is likely to be a compensation effect for the loss of CD11b+Ly6G+Ly6C+ cells or an inflammatory reaction to the minor toxicity caused by the DT. We have added the explanation in the discussion section (line 340-342)

3.) Fig 2B clearly shows that tumor-bearing mice die significantly earlier (p<0.02) when DT is applied. However, the authors claim that this effect is not significant when more mice are pooled for analysis and refer to Fig S3D at that point. Why do the author not show the whole number of mice analyzed in Fig. 2B, but only a subset?

It is also not clear how Fig S3D relates to Fig 2B as no DT-curves are shown in comparison to PBS-curves in Fig S3D as they are presented in Fig 2B. Instead GCV curves are compared to PBS curves. Thus, the data in Fig S3D do not appear to support the claim that “this effect is not significant when more mice are pooled”. Please clarify this and clearly show only relevant data supporting your claim. 

Response 3:  We are sorry for the inaccurate message.  These were all mice we have for this experiment.  Despite the statistical difference, we did not feel so comfortable for the surviving data (one criterion for the surviving is the change of body weight) because we found that the mice treated by DT are weaker than normal mice. This is why we used the tumor size to re-verify the effect on tumor growth (Fig. 2C and 2D). We have re-phrased the sentences (line 209-211). I hope it is much clear now.

4.) The authors say that “DT treatment slightly reduced the total amount of CD11b cells myeloid cells” which is visible in Fig. 3A but presented as not significant in Fig. 3B. Can the authors increase the number of samples tested (higher n) to reach significance?

Response 4: In this transgenic mouse model, the primary cell-depleted by our protocol is F4/80+ cells that are about 50% of CD11b+ cells. It will need more mice to reach significance for CD11b+ cells.   The DTR transgenic mice were not available currently.  We need to breed a new batch of mice and re-submit the application for the approval of animal use. This will take a few months to increase the number of samples tested.  We, therefore, hesitate to perform this experiment and prefer to leave it in this way.  

5.) Please introduce and explain the tk/GCV therapy system. No reader who is not a particular expert in your field knows what it is. It is impossible to evaluate the data, if it is not even clear how the main experimental system works. It has to be explained in the main text, Fig S3 is not sufficient for that.

Response 5:  The introduction of tk/GCV has been added to the introduction section (line 73-83).

6.) In the discussion the authors mention that clinical inhibition of CSF1R and/or CCL2 is our investigation in other tumor types. Is it possible to test inhibitors of these targets in the here employed mouse model to confirm the findings of this study using another method and to present a more specific therapeutic indication form glioma treatment? 

Response 6: I afraid that we cannot perform this experiment at this moment. Several issues need to be considered. The first is that the Animal Ethics Committee Approval protocol that we used in this study did not include this experimental design. If we want to perform this experiment, a new approval is required, which will take time and also another few months to repeat the experiment that has been published in other tumor models. The second is the budget issue for performing a new experiment, which was not included in our original grant budget. We need to find a new budget source for this experiment. We will, therefore, prefer to cite references in supporting our statement (ref 65-67, line 364).

Minor remarks:

1.) The abbreviation GCV is explained or introduced at no point in the manuscript. Please introduce this compound to the readers.

Response 7:  The abbreviation of GCV has been added and explained in the introduction section (line77 ).

2.) Is it necessary to show in Fig. 4D that sick mice of both populations have similar tumor sizes?

Response 8:  The reason to show that sick mice of both populations have similar tumor sizes is to indicate that mice mainly died from the tumor pressure, not other reasons. Although this is a piece of indirect evidence, we think it is not a distraction for the readers. However, if the reviewer considers it may lead in the wrong direction, we can take it out.

3.) Text within the figures is frequently too small to read in a printout, e.g. Figure 5B.

Response 9:  The text font within the figures has been changed to a larger font.

Please also see attached file

Reviewer 2 Report

In this study, Sheng-Yan Wu and Chi-Shiun Chiang investigated the role of a subpopulation of myeloid derived cells, characterized as CD11b positive, Ly6G negative and Ly6C negative cells, in brain tumor progression in absence and in presence of drug treatment. For that purpose, they used an orthotopic murine model that consists in the implantation of a murine astrocytoma cell line they established into the brain of CD11b-diphtheria toxin (DT) receptor transgenic mice. This conditional Knock-out model has been extensively used to study the role of CD11b positive cells and is well validated. The authors had observed (most likely unpublished experiments?) that the main population of blood cells depleted in these mice were CD11b+Gr1- cells, but not the CD11b+Gr1+ cells. As stated by the authors in the abstract, the CD11b+Ly6G+Ly6C+ granulocytic monocyte-derived suppressor cells (G-MDSCs) and the CD11b+Ly6G-Ly6C+ monocytic MDSCs (M-MDSCs) have been extensively studied in cancer biology because of their immunosuppressive activities. Much less is known about the CD11b+Ly6G-Ly6C- cells they analyze in this report.

Following two injections of DT in the mice, they observed and hence confirmed a decrease in the amount of CD11b+Ly6G-Ly6C- present in the blood, as analyzed by flow cytometry. Interestingly, the decrease in CD11b+Ly6G-Ly6C- cells appears to be transient, as a percentage of cells similar to that observed at day 0 is reported at day 6 (Figs 1 and 1S). In the brain tumor model, induction of CD11b+ cells depletion by DT injection led to a decrease in animal survival and affected neither the tumor volume nor the tumor growth kinetics as compared to control mice. Analysis by immunohistochemistry of brain sections showed a decrease in signals for CD11b and F4/80 but not for Gr1 after DT injection. The authors interpret this as a decrease of CD11b+Ly6G-Ly6C- and F4/80 tumor-associated macrophages (TAM). However, when a treatment such as Ganciclovir is applied at the time the depletion is induced, a significant improvement in animal survival was observed, parallel to a decrease in CD11b+Ly6G-Ly6C- cells and F4/80 tumor-associated macrophages (TAM). This is no longer the case if depletion and treatment are uncoupled.

This study contributes interesting data about the CD11b+Ly6G-Ly6C- cells and their manipulation in treated tumors, which might be of relevance for the design of future therapies. It however requires and deserves some improvement.

My first recommendation, as a non-native English speaker myself, is to have the manuscript read and edited by a native English speaker for the grammar and the syntax.

General comments:

ABSTRACT

-check that every abbreviation is defined (e.g. GCV, TAM)

INTRODUCTION

-lines 37-39: the authors start to talk about glioma then they switch to GBM. The GBM abbreviation is not defined. Please give the full name for GBM before introducing the abbreviation. Make clear in the text whether you are talking of glioma in general or more specifically of glioblastoma.

-line 47: “noticed” or “notice” ?

-line 48: “sharing” or “share”?

-lines 56-58: please cite a reference to support the statement that “…CD11b+Ly6G-Ly6C- myeloid-derived cells (MDCs) in the blood were less discussed despite that they were generally regarded as the precursor cells of MDSCs or macrophages”.

-line 68: is the reference 24 appropriate ? Or would it be better to add a review on the GBM-associated microglia/macrophages?

-lines 86-87: the authors speak about an initial study with the DTR-mice. If the observations have been published, the reference should be added. If not, I would recommend to add the word “preliminary” or “ not published”. Moreover, they now speak of CD11b+Gr1+ or Gr1- cells. Later in the manuscript, they never refer again to the Gr1 molecule. For readers who are not familiar with the complex world of myeloid cells, it is difficult to understand why the authors speak of Gr1 or Ly6. Since the journal Cells is not a journal targeting only immunologists, it is important to give the reader some background on the definition at the molecular level of the myeloid cells.

RESULTS

-Figure 1B: it is impossible to distinguish the bars used for the legend. Maybe use another filling?

-line 190: the authors write: “The red blood cells lysed blood cells...”. This is not really easy to understand! The authors often use this phrasing (e.g. in the legend of the Supp Fig 1: CD11b+ blood lysed cells). This must be checked and edited throughout the text.

-Figs 1C and Supp2A: I see a transient decrease in the amount of CD11b+Ly6G-Ly6C- cells: less cells at day 3, more at day 6, right? The authors should comment on this point.

-Figure 3:

How was the quantification of the labelling performed? How was the vessel density measured and quantitated? These should be described and added in the Methods part.

How do we know that we are looking at the tumor region? I would recommend to add an image showing the difference between “tumor-free brain” and “tumor-harboring brain” regions.

-line 256: do the authors mean macrophages only or all the myeloid cells they analyze?

-Figure 5F: the effect of temozolomide is reported in the manuscript only for the tumor size. Could the authors comments on the survival of the animals?

-Figure 6, table: I would recommend to add “GCV” in the “only” cell

-Analysis of the myeloid cell subpopulations: the authors have used flow cytometry to analyze blood cells and peritoneal cells and IHC to analyze the brain. Flow cytometry could have been used as well for the brain tumors. It would have brought more qualitative data. Or did the authors use IHC because of a quantitative issue (not enough CD11b+Ly6G-Ly6C- cells to get a good analysis)?

DISCUSSION:

-line 331: the word “fabulous” is maybe a bit too strong

-lines 356-357: the authors state that: “effectiveness of TAMs depletion could only be seen when it was performed during therapeutic period, not before or after the therapy, neither during tumor growing period.” The first and last part of the sentence are supported by their data, but not the middle part that reads: “not before or after the therapy”. I did not find the data related to this statement in the manuscript. Please clarify this issue and correct the text if necessary.

-I am missing a discussion on the possible translation of these observations to therapy.

REFERENCES:

-line 448: please check the reference 10: it seems that the citation is not correctly written.

Author Response

Response to Reviewer 2 comments

My first recommendation, as a non-native English speaker myself, is to have the manuscript read and edited by a native English speaker for the grammar and the syntax.

Response 1: Prof. Willam H. McBride from UCLA has proofread the manuscript.  We have added his contribution to the acknowledgment section. Many incorrect sentences have been rewritten. Hopefully, the clarity of this manuscript has been improved.

General comments:

ABSTRACT

-check that every abbreviation is defined (e.g. GCV, TAM)

Response 2: All abbreviations are defined.

INTRODUCTION

-lines 37-39: the authors start to talk about glioma then they switch to GBM. The GBM abbreviation is not defined. Please give the full name for GBM before introducing the abbreviation. Make clear in the text whether you are talking of glioma in general or more specifically of glioblastoma.

Response 3: Necessary changes have been made.

-line 47: “noticed” or “notice” ?

Response 4: Sentence has been rewritten.

-line 48: “sharing” or “share”?

Response 5: Sentence has been rewritten.

-lines 56-58: please cite a reference to support the statement that “…CD11b+Ly6G-Ly6C- myeloid-derived cells (MDCs) in the blood were less discussed despite that they were generally regarded as the precursor cells of MDSCs or macrophages”.

Response 6: References (ref. 10 and 15) are cited.

-line 68: is the reference 24 appropriate ? Or would it be better to add a review on the GBM-associated microglia/macrophages?

Response 7: Reference 24 has been replaced with a new reference.

-lines 86-87: the authors speak about an initial study with the DTR-mice. If the observations have been published, the reference should be added. If not, I would recommend to add the word “preliminary” or “ not published”. Moreover, they now speak of CD11b+Gr1+ or Gr1- cells. Later in the manuscript, they never refer again to the Gr1 molecule. For readers who are not familiar with the complex world of myeloid cells, it is difficult to understand why the authors speak of Gr1 or Ly6. Since the journal Cells is not a journal targeting only immunologists, it is important to give the reader some background on the definition at the molecular level of the myeloid cells.

Response 8: We have changed to “our study” and rewritten the sentence (line 72-74). We have also added the definition of various subtypes of myeloid cells in the introduction section (line 47-49).

RESULTS

-Figure 1B: it is impossible to distinguish the bars used for the legend. Maybe use another filling?

Response 9:  The bars within the figures have been changed to improve the contrast.

-line 190: the authors write: “The red blood cells lysed blood cells...”. This is not really easy to understand! The authors often use this phrasing (e.g. in the legend of the Supp Fig 1: CD11b+ blood lysed cells). This must be checked and edited throughout the text.

-Figs 1C and Supp2A: I see a transient decrease in the amount of CD11b+Ly6G-Ly6C- cells: less cells at day 3, more at day 6, right? The authors should comment on this point.

Response 10:  The phrase has been changed to “red cell lysed blood” (line 183).

-Figure 3:

How was the quantification of the labelling performed? How was the vessel density measured and quantitated? These should be described and added in the Methods part.

Response 11:  The quantification of the MVD and IHC statistics has been added in the method section (line 163-164)

How do we know that we are looking at the tumor region? I would recommend to add an image showing the difference between “tumor-free brain” and “tumor-harboring brain” regions.

Response 12:  The tumor region is easily recognized by the denser nuclei staining, either by DAPI or H&E reagent, as one example in Fig. 5G. We have added it in the material section (line 161-162)

-line 256: do the authors mean macrophages only or all the myeloid cells they analyze?

Response 13:  We have rewritten the sentences to make it clear that we analyzed all the myeloid cells. Only the group of cells with CD11b+F4/80+ is macrophages.

-Figure 5F: the effect of temozolomide is reported in the manuscript only for the tumor size. Could the authors comments on the survival of the animals?

Response 14: The mice receiving both TMZ and DC treatments were too sick to survive longer than 20 days. It is, therefore, impossible to analyze the survival of the animals as the result of tumor burden rather than the toxicity of TMZ + DC. We have added the explanation in the result section (line 274-276).

-Figure 6, table: I would recommend to add “GCV” in the “only” cell

Response 15: It has been changed.

-Analysis of the myeloid cell subpopulations: the authors have used flow cytometry to analyze blood cells and peritoneal cells and IHC to analyze the brain. Flow cytometry could have been used as well for brain tumors. It would have brought more qualitative data. Or did the authors use IHC because of a quantitative issue (not enough CD11b+Ly6G-Ly6C- cells to get a good analysis)?

Response 16:  Yes, the prime reason that we used IHC for brain tumors is the cell number. We have tried to set up the system to monitor the myeloid cells in the brain tumor. However, the tumor in the brain was too small that we need to pull brains from several mice together to get one data. It is even harder to collect the treated tumor because the tumor was much smaller than control.

DISCUSSION:

-line 331: the word “fabulous” is maybe a bit too strong

Response 17:  The sentence has been rewritten.

-lines 356-357: the authors state that: “effectiveness of TAMs depletion could only be seen when it was performed during therapeutic period, not before or after the therapy, neither during tumor growing period.” The first and last part of the sentence are supported by their data, but not the middle part that reads: “not before or after the therapy”. I did not find the data related to this statement in the manuscript. Please clarify this issue and correct the text if necessary.

Response 18:  We redrawed Fig. 6 A to give a much clear presentation for the administration before (day 8 and 9) or after (day 16 and 17) the therapy (day 10 to 15) using pre-GCV or post-GCV, respectively.  We have re-phrased the figure legend to make it clear for the “before or after” treatment.  We also clarify this issue in the result section.

-I am missing a discussion on the possible translation of these observations to therapy.

Response 19:  We have added sentences to discuss the possible translation of this study. (line 364-365)

REFERENCES:

-line 448: please check the reference 10: it seems that the citation is not correctly written.

Response 20:  Error has been fixed.

Please also see attached file

Round 2

Reviewer 1 Report

Revision round two: Major remarks: Point 1) English language is strongly improved and acceptable this way. Point 2) The explanation is plausible. Point 3) The explanation is sufficient. Point 4) As Figure 3B represents a minor finding, it can stay this way. Point 5) The introduction of the experimental model is sufficient. Minor remarks: Point 1) Ok Point 2) Please place your explanation that mice died from tumor pressure, not for other reasons, in the main text or the figure legend of Fig. 4D. Point 3) Ok Recommendation: Accept with minor revision.

Author Response

Point 2) Please place your explanation that mice died from tumor pressure, not for other reasons, in the main text or the figure legend of Fig. 4D.

Response 5: We have put the explanation in the main text section (line 252-254).